# Microbial Pathway Thermodynamics: Stoichiometric Models Unveil Anabolic and Catabolic Processes

**DOI:** 10.3390/life14020247

**Published:** 2024-02-09

**Authors:** Oliver Ebenhöh, Josha Ebeling, Ronja Meyer, Fabian Pohlkotte, Tim Nies

**Affiliations:** 1Institute of Quantitative and Theoretical Biology, Heinrich Heine University Düsseldorf, 40225 Düsseldorf, Germany; 2Cluster of Excellence on Plant Sciences, Heinrich Heine University Düsseldorf, 40225 Düsseldorf, Germany

**Keywords:** energy metabolism, elementary conversion modes, metabolic networks, energy converter

## Abstract

The biotechnological exploitation of microorganisms enables the use of metabolism for the production of economically valuable substances, such as drugs or food. It is, thus, unsurprising that the investigation of microbial metabolism and its regulation has been an active research field for many decades. As a result, several theories and techniques were developed that allow for the prediction of metabolic fluxes and yields as biotechnologically relevant output parameters. One important approach is to derive macrochemical equations that describe the overall metabolic conversion of an organism and basically treat microbial metabolism as a black box. The opposite approach is to include all known metabolic reactions of an organism to assemble a genome-scale metabolic model. Interestingly, both approaches are rather successful at characterizing and predicting the expected product yield. Over the years, macrochemical equations especially have been extensively characterized in terms of their thermodynamic properties. However, a common challenge when characterizing microbial metabolism by a single equation is to split this equation into two, describing the two modes of metabolism, anabolism and catabolism. Here, we present strategies to systematically identify separate equations for anabolism and catabolism. Based on metabolic models, we systematically identify all theoretically possible catabolic routes and determine their thermodynamic efficiency. We then show how anabolic routes can be derived, and we use these to approximate biomass yield. Finally, we challenge the view of metabolism as a linear energy converter, in which the free energy gradient of catabolism drives the anabolic reactions.

## 1. Introduction

Microbial organisms are an essential part of most ecosystems. They function as vital members of natural production chains, leading to the formation of chemical compounds that have complexity which is unreachable by current technological standards [1]. Therefore, much scientific effort has been spent to understand the metabolism of microbes [2]. Today, the human exploitation of microbial metabolism has long left the stage of merely producing fermented food, such as cheese or alcoholic beverages [3]. Microbes, often viewed as natural factories, are used in biotechnological applications as an integrated component of drug and food fabrication or bioremediation projects [4,5,6]. In order to assess the potential of a microbial organism for bioeconomical applications, it is important to fully understand its metabolic capabilities. This includes a thermodynamic characterization of metabolic pathways, which allows us, for example, to calculate maximally possible yields.

Gaining full knowledge about microbial metabolism was and is a complex scientific problem. This complexity derives from the multitude of microbial organisms, the diversity of metabolic pathways, and the different strategies to regulate metabolism [7,8,9]. In the pre-genomic era, researchers used precise measurements of input (substrate) and output (product) relationships for microbial cultures and a deep understanding of thermodynamics to design new biotechnological strategies, based on so-called macrochemical equations [2,10,11,12,13,14]. A macrochemical equation summarizes the conversion of substrates into metabolic products and biomass. Describing microbial metabolism by a single macrochemical equation essentially treats microbial metabolism as a black box, ignoring all intracellular metabolic details. Still, this single chemical equation can accurately describe the overall metabolic activity and thus can serve as a means to understand and predict metabolic properties (see, e.g., [11]).

The macrochemical equation can be understood as a sum of two separate reactions which describe catabolism (the breakdown of nutrients to gain free energy in the form of ATP) and anabolism (the formation of new biomass from the nutrients using the energy in the form of ATP provided by catabolism) [15]. In this picture, microbial growth is described as a thermodynamic energy converter, where the catabolic reactions provide the required free energy to drive anabolism (see Figure 1). Here, the negative free energies of reactions of the catabolic and anabolic half-reactions, denoted by $-\Delta_{\mathrm{cat}}G$ and $-\Delta_{\mathrm{ana}}G$, respectively, are generalized thermodynamic forces, and the respective reaction rates, $J_{\mathrm{cat}}$ and $J_{\mathrm{ana}}$, are generalized thermodynamic fluxes. In the theory of non-equilibrium thermodynamics, it has been shown that in general, for systems close to equilibrium, the generalized fluxes *J* depend linearly on generalized forces *X* (*X* is a vector combining $-\Delta_{\mathrm{cat}}G$ and $-\Delta_{\mathrm{ana}}G$), i.e., J=LX, where the entries $L_{ij}$ of the matrix L are the so-called phenomenological coefficients [16]. The name phenomenological coefficient refers to the fact that the quantities cannot be derived from first principles but they have to be experimentally measured. It is often assumed that the linear relationship is also valid for catabolic and anabolic forces and fluxes [14,17,18,19], which would entail that the following relation holds:(1)$$\left(\begin{array}{c} J_{\mathrm{cat}}\\ J_{\mathrm{ana}} \end{array}\right)=L\;\cdot \;\left(\begin{array}{c} -\Delta_{\mathrm{cat}}G\\ -\Delta_{\mathrm{ana}}G \end{array}\right).$$

However, it is thus far not clear why this actually should be the case, considering that metabolically active and dividing cells operate far from thermodynamic equilibrium. We test the assumption of a linear relationship with published experimental data for *S. cerevisiae* and *E. coli* grown in chemostat cultures and observe that the measured data clearly contradict this assumption.

The experimental determination of a macrochemical growth equation requires precise measurements of all chemical substances that are consumed and produced by the growing microbes. These measurements are possible in controlled chemostat cultures, in which microbes grow on a defined growth medium. However, many microbes cannot easily be cultured in chemically defined media, which prevents an experimental determination of macrochemical growth equations. With the recent scientific advancement in genome sequencing, genomic data became available for a huge number of organisms, including those which are difficult to culture [20]. This information allows for building stoichiometric (genome-scale) metabolic models, manually or semi-automatically [21,22,23,24]. Genome-scale metabolic models are a formalization of all known biochemical reactions of an organism. As such, they combine genomic, proteomic, and metabolic information to build an in silico representation that can be used to derive steady-state flux distributions [25]. The latter is achieved by the optimization of an objective function (such as biomass production) using linear programming. The inspection and analysis of genome-scale metabolic models benefit from a rich theory for metabolic networks (see, e.g., [25,26,27,28,29]). Thus, such models can support strategies to improve the product yield in biotechnological applications. Elementary flux modes (EFMs) are a systematic way to quantify the metabolic capabilities of an organism [26]. EFMs describe all possible pathways between substrate and product. Due to combinatorial explosion [30], it is still challenging to calculate all EFMs for larger structural metabolic models, also with modern computational facilities. To overcome this, and because for many investigations only the conversion between substrate and product is relevant, elementary conversion modes (ECMs) were introduced by Urbanczik and Wagner [31]. According to [32], elementary conversion modes are the minimal building blocks of all net conversions (input–output relationships) in a metabolic network. Plainly speaking, ECMs ignore all intracellular processes and only focus on the effects of metabolic pathways on the external metabolites. Using ECMs instead of EFMs reduces the necessary computational power drastically. Additionally, modern software, such as ecmtool (https://github.com/SystemsBioinformatics/ecmtool, last accessed 29 November 2023) which allows for the parallelization of the computation, helps us to obtain an exhaustive list of all metabolic capabilities of an organism in the form of ECMs [33].

As suggested in [32], we view ECMs as building blocks of macrochemical equations. We show how genome-scale metabolic models can be used to systematically enumerate all possible catabolic pathways. With thermodynamic data, in particular the energies of formation of substrates and products obtained from the eQuilibrator tool [34,35], we characterize the catabolic pathways by their energy gradient, by which we understand the free energy that becomes available from converting nutrients with a high energy of formation into catabolic products with a lower energy of formation. Using the network models, we further estimate the maximal ATP production capacity for each catabolic pathway, and we thus determine their thermodynamic efficiencies. We then analyze experimental data for *Saccharomyces cerevisiae* [36] and *Escherichia coli* [37], grown under controlled chemostat conditions in defined media, to separate the macrochemical equation into its catabolic and anabolic parts and characterize their thermodynamic properties. Interpreting our findings in the context of the energy converter model, we identify the limitations of the applicability of the linear converter model, but we observe an interesting linear scaling law between growth rate and metabolic power.

## 2. Theory and Methods

### 2.1. Calculating Elementary Conversion Modes

Elementary conversion modes (ECMs, [31]) are a concept related to elementary flux modes (EFMs, [26]). However, whereas an EFM describes a minimal set of reactions in a network which can carry a stationary flux through the network, ECMs are not defined in terms of the metabolic routes through the network but by their end results, i.e., by the stoichiometry in which they convert external metabolites. As such, enumerating all ECMs of a network provides a full description of all possible metabolic conversions by the network. Here, we employ ECMs to thermodynamically characterize catabolic routes. For this, we calculate ECMs which connect external metabolites but which do not lead to the production of biomass. We perform calculations for three metabolic models, the *E. coli* core model [38], the genome-scale model iJR904 for *E. coli* str. K-12 substr. MG1655 [39], and the genome-scale model iND750 for the yeast *S. cerevisiae* [40], using ecmtool [32,33]. Due to their size, for the latter two (genome-scale) models, a complete enumeration of all ECMs is not easily possible with standard computing equipment. We therefore hid all external metabolites that contain phosphate, sulfur, or nitrogen and dismissed all compounds with more than six carbon atoms. These steps reduced the number of catabolic routes considerably and allowed the calculations to be performed in a reasonable time. As input (carbon source) for ecmtool, we focused on simple sugars and carboxylic acids (glucose, xylose, pyruvate, 2-oxoglutarate). Moreover, we allowed oxygen to be present. The output of ecmtool is a matrix, in which the rows are the respective elementary conversion modes and the columns are all external metabolites that were not hidden. Excluding larger molecules and those containing sulfur, phosphorus, and nitrogen reduces the number of conversion modes and thus the number of possible catabolic routes. For example, ecologically important catabolic pathways obtaining free energy from oxidizing ammonia [41] are excluded by our simplification. For the *E. coli* core network [38], no metabolite had to be hidden, and thus the full catabolic potential of the core network could be described.

### 2.2. Estimating Gibbs Free Energy of Reaction and Thermodynamic Efficiency

To approximate the standard Gibbs free energy of reaction ($\Delta_{\mathrm{cat}}G^{\circ }$) for each obtained ECM, we used the Python API of the eQuilibrator tool [35]. We extracted the Gibbs free energies of formation ($\Delta_{f}G^{\circ }$) for all external metabolites involved in an ECM. Next, we normalized the ECMs with respect to the carbon atoms of the carbon source (C-mol) and applied a Laplace transformation, adapting for temperature (298.15 K), pH (7.4), pMg (3.0), and ionic strength (0.25 M). We used Hess’s law to calculate the standard Gibbs free energy of reaction for each ECM,
(2)$$\Delta_{\mathrm{cat}}G^{\circ }=\sum_{i=1}^{m}\nu_{i}\Delta_{f}G_{i}^{\circ },$$
where $\nu_{i}$ and $\Delta_{f}G_{i}^{\circ }$ are the stoichiometric coefficient and the Gibbs free energy of formation of the $i^{\mathrm{th}}$ external compound in the ECM, respectively.

In the following, we will use the standard Gibbs free energies as an objective quantity to characterize a large number of catabolic routes. It has to be noted, though, that the actual free energy gradients depend on the concentrations of substrates and products in the solution and that in physiological conditions cells never experience standard conditions. If all concentrations $c_{i}$ of the external compounds are known, then the actual energy of reaction can be determined by
(3)$$\Delta_{\mathrm{cat}}G=\Delta_{\mathrm{cat}}G^{\circ }+RT\ln\prod_{i=1}^{m}c_{i}^{\nu_{i}}.$$

Whereas for single biochemical reactions the concentration-dependent term in Equation (Equation 3) is often of the same order of magnitude as the standard energy of reaction and can therefore critically affect the thermodynamics and directionality of a reaction, its relative importance is much lower for overall pathways, which consist of several reactions and thus exhibit a larger total energy of reaction. The pathways considered here display standard energies of reaction between 30 and 500 kJ ${\text{C-mol}}^{-1}$, which corresponds to between 180 and 3000 $kJ\;{\mathrm{mol}}^{-1}$ substrate. On the other hand, for each order-of-magnitude deviation from standard conditions, the second term results in a correction of only $RT\ln10=5.7kJ\;{\mathrm{mol}}^{-1}$. Moreover, the estimations of the standard energies of reaction, which are based on the group contribution method [42], come with a considerable margin of error. For example, assuming typical glucose concentrations for substrate unlimited growth of $c_{\mathrm{Glc}}=0.16\mathrm{mol}\;L^{-1}$ ($\approx 30\;g\;L^{-1}$), dissolved oxygen of $c_{O_{2}}=250\;\mu \mathrm{mol}\;L^{-1}$, and carbon dioxide of $c_{{\mathrm{CO}}_{2}}=13\;\mu \mathrm{mol}\;L^{-1}$ (corresponding to ambient air with 20% oxygen and 420 ppm carbon dioxide), the second term computes to $-39.4\;kJ\;{\mathrm{mol}}^{-1}$ (at 25${\;}^{\circ }C$), in which (i) still lies well within the error range of the computational prediction of the standard energy of reaction, $\Delta_{\mathrm{cat}}G^{\circ }=-2927.8\pm 49.5\;kJ\;{\mathrm{mol}}^{-1}$ [34], and (ii) is two orders of magnitude smaller than the standard energy of reaction itself. Taking into account the total energy gradients and the errors in the prediction method of reaction energies, we can expect that the standard energies of reaction provide a sufficiently good thermodynamic characterization of catabolic pathways.

For the calculation of the maximal ATP production for an ECM, we constrained all external fluxes to the values of the respective ECMs while maximizing ATP hydrolysis (excluding ATP maintenance):(4)$$\begin{array}{c} \begin{array}{cc} \mathrm{maximize}\; & v_{\mathrm{ATPM}},\\ \mathrm{such}\;\mathrm{that}\; & N\;\cdot \;v=0\\ \; & v_{i,\mathrm{ex}}=\nu_{i}\;\mathrm{for}\;i\in \mathrm{ECM},\\ \; & v_{j,\mathrm{ex}}=0\;\mathrm{for}\;j\notin \mathrm{ECM}, \end{array} \end{array}$$
where N is the stoichiometric matrix of the metabolic model, $v_{\mathrm{ATPM}}$ is the flux through the reaction
(5)$$\mathrm{ATP}+H_{2}O⇌\mathrm{ADP}+P_{i},$$
and $v_{i,\mathrm{ex}}$ are the fluxes through the reaction exchanging metabolite *i*, which is constrained to the stoichiometric coefficient $\nu_{i}$ obtained by the respective ECM. The stoichiometric coefficients are normalized to one carbon mole substrate.

The thermodynamic efficiency of ATP production is calculated as
(6)$$\eta =\frac{c_{\mathrm{ATP}}\;\cdot \;\Delta_{r}G_{\mathrm{ATPase}}}{|\Delta_{\mathrm{cat}}G|},$$
where $c_{\mathrm{ATP}}$ is the maximal ATP yield per carbon mole, $\Delta_{r}G_{\mathrm{ATPase}}$ is the energy of reaction for ATP synthesis from ADP and inorganic phosphate, and $\Delta_{\mathrm{cat}}G$ is the energy of reaction of the respective catabolic pathway.

For the Gibbs free energy of ATP synthesis, we used a typical value for *E. coli* of $\Delta_{r}G_{\mathrm{ATPase}}=46.5\;kJ\;{\mathrm{mol}}^{-1}$ [43].

### 2.3. Calculating the Stoichiometry of Anabolism

We assume that the substrate [S] has the normalized sum formula $\begin{array}{c} {\mathrm{CH}}_{x}O_{y} \end{array}$, the biomass [X] has $\begin{array}{c} {\mathrm{CH}}_{a}O_{b}N_{c} \end{array}$, and that the biomass is more reduced than the substrate, i.e.,
(7)$$\gamma_{S}=4+x-2y\leq \gamma_{X}=4+a-2b-3c.$$

We assume an overall stoichiometry of
(8)$$b_{1}\;\left[S\right]+b_{2}\;{\mathrm{NH}}_{3}⟶\left[X\right]+b_{3}\;{\mathrm{CO}}_{2}+b_{4}\;H_{2}O.$$

Every carbon that is converted to biomass will have to be reduced by $\gamma_{X}-\gamma_{S}$. From the overall redox balance, it follows that for each carbon that is converted into biomass,
(9)$$b_{3}=\frac{\gamma_{X}-\gamma_{S}}{\gamma_{S}}$$
carbons have to be oxidized to CO_2_. From the carbon balance of (Equation 8), it follows that
(10)$$b_{1}=1+b_{3}=\frac{1}{Y_{X/S}^{\max}}.$$

The nitrogen and hydrogen balances entail that
(11)$$b_{2}=b\;\mathrm{and}\;b_{4}=\frac{b_{1}x+3b_{2}-a}{2}.$$

It is straight-forward to generalize these calculations to include sulfur and phosphorus into the biomass.

### 2.4. Calculating the Optimal Anabolic Reaction from Genome-Scale Networks

To determine the maximal yield and the minimal ATP requirement for biomass formation, we perform two subsequent linear programs. First, the exchange reactions are constrained, such that only the carbon source (substrate) and oxygen can be imported (negative flux), but other metabolites can be released (positive flux). The biomass reaction is constrained to one carbon mole per unit of time. The ATP hydrolysis reaction is not constrained, which means it can run in reverse and provide ATP. Subsequently, substrate import (negative) is maximized: (12)$$\begin{array}{c} \begin{array}{cc} \mathrm{maximize}\; & v_{\mathrm{substrate},\mathrm{ex}},\\ \mathrm{such}\;\mathrm{that}\; & N\;\cdot \;v=0\\ \; & v_{\mathrm{biomass}}=1,\\ \; & -\infty <v_{\mathrm{substrate},\mathrm{ex}}\leq 0\\ \; & -\infty <v_{O_{2},\mathrm{ex}}\leq 0\\ \; & 0\leq v_{j,\mathrm{ex}}<\infty \;\mathrm{for}\;\mathrm{other}\;\mathrm{metabolites} \end{array} \end{array}$$

The resulting optimal flux is negative, and the absolute value denotes the minimal substrate requirement to produce one carbon mole of biomass, if ATP is provided in abundance.

In the second step, the determined minimal substrate requirement $v_{\mathrm{substrate}}^{\mathrm{opt}}$ is fixed, and the ATP requirement is minimized by maximizing the (negative) flux through reaction (Equation 5): (13)$$\begin{array}{c} \begin{array}{cc} \mathrm{maximize}\; & v_{\mathrm{ATPM}},\\ \mathrm{such}\;\mathrm{that}\; & N\;\cdot \;v=0\\ \; & v_{\mathrm{biomass}}=1,\\ \; & v_{\mathrm{substrate},\mathrm{ex}}=v_{\mathrm{substrate}}^{\mathrm{opt}}\\ \; & -\infty <v_{O_{2},\mathrm{ex}}\leq 0\\ \; & 0\leq v_{j,\mathrm{ex}}<\infty \;\mathrm{for}\;\mathrm{other}\;\mathrm{metabolites} \end{array} \end{array}$$

The absolute value of the optimal flux $|v_{\mathrm{ATPM}}|$ gives a minimal ATP requirement for the production of one carbon mole biomass.

### 2.5. Calculating the Stoichiometry of Catabolism

Macrochemical equations of the form
(14)$$\begin{array}{cc} \; & \left[S\right]+\alpha_{1}\;O_{2}+c\alpha_{2}\;{\mathrm{NH}}_{3}⟶\\ \; & \;\alpha_{2}\;\left[X\right]+\alpha_{3}\;C_{2}H_{5}\mathrm{OH}+\alpha_{4}\;{\mathrm{CO}}_{2}+\alpha_{5}\;H_{2}O, \end{array}$$
for *S. cerevisiae* (see [36]) and
(15)$$\begin{array}{cc} \; & \left[S\right]+\alpha_{1}\;O_{2}+c\alpha_{2}\;{\mathrm{NH}}_{3}⟶\\ \; & \;\alpha_{2}\;\left[X\right]+\alpha_{3}\;{\mathrm{CH}}_{3}\mathrm{COOH}+\alpha_{4}\;{\mathrm{CO}}_{2}+\alpha_{5}\;H_{2}O, \end{array}$$
for *E. coli* were obtained from the original publications. Here, we use the notation [S] for one carbon mole of substrate and [X] for one carbon mole of biomass. The sum formula of biomass is assumed to be given by $\begin{array}{c} {\mathrm{CH}}_{a}O_{b}N_{c} \end{array}$ (hence the factor *c* in the stoichiometry of NH_3_), and it is given in both cases in the original publication. The stoichiometric coefficients were obtained as follows. For *S. cerevisiae*, Table 1 in [36] already provides the stoichiometric coefficients for Equation (Equation 22), which were, for our calculations, converted into carbon moles. For *E. coli*, we converted data from Table 2 in [37] which are given in $g\;g^{-1}\;h^{-1}$ to C-mol ${\text{C-mol}}^{-1}$ $h^{-1}$, using the molecular weights of the chemical compounds as well as the biomass, normalized to one carbon mole.

The coefficients for the catabolic reaction
(16)$$c_{1}\;\left[S\right]+c_{2}\;O_{2}⟶c_{3}\;{\mathrm{CO}}_{2}+c_{4}\;C_{2}H_{5}\mathrm{OH}+c_{5}\;H_{2}O$$
are simply determined by calculating (Equation 22) $-\alpha_{2}\;\cdot \;$ (Equation 8), resulting in the coefficients
(17)$$\begin{array}{cc} c_{1} & =1-b_{1}\alpha_{2} \end{array}$$
(18)$$\begin{array}{cc} c_{2} & =\alpha_{1} \end{array}$$
(19)$$\begin{array}{cc} c_{3} & =\alpha_{4}-b_{3}\alpha_{2} \end{array}$$
(20)$$\begin{array}{cc} c_{4} & =\alpha_{3} \end{array}$$
(21)$$\begin{array}{cc} c_{5} & =\alpha_{5}-b_{4}\alpha_{2}. \end{array}$$

Subsequently, it is convenient to normalize this equation to the consumption of one carbon mole of substrate, i.e., dividing all coefficients by $c_{1}$.

In the case of *E. coli*, acetate was excreted instead of ethanol at the onset of overflow metabolism [37]. In the calculation, ethanol can simply be replaced by acetic acid and the calculation remains identical.

### 2.6. Extraction and Processing of Experimental Data

For the experimental determination of anabolic and catabolic fluxes, it is necessary to precisely measure the macrochemical growth equation. Chemostat cultures are best suited for such measurements, because growth rates can be precisely controlled by setting the dilution rate, and cultures will approach a stationary growth. Basically, all input and output fluxes need to be quantified. This entails that nutrient concentrations in the feed and in the culture vessel need to be determined. Moreover, gas exchange rates for oxygen and CO_2_ need to be measured. Further, ideally, all metabolic products should be quantified. A commonly applied criterion to check whether all products have been measured is the so-called carbon recovery coefficient, which quantifies the fraction of carbons that were provided with the nutrients, which are recovered in the biomass and the metabolic products.

Specifically, we obtain data from two sources. For *S. cerevisiae*, the data source is Table 1 in [36], which provides the stoichiometric coefficients of the macrochemical equation in the form
(22)$$\begin{array}{cc} \; & C_{6}H_{12}O_{6}+\alpha_{1}\;O_{2}+0.15\alpha_{2}\;{\mathrm{NH}}_{3}⟶\\ \; & \;\alpha_{2}\;CH_{1.79}O_{0.57}N_{0.15}+\alpha_{3}\;C_{2}H_{6}O+\alpha_{4}\;{\mathrm{CO}}_{2}+\alpha_{5}\;H_{2}O. \end{array}$$

Here, we convert these coefficients to match the macrochemical equation in the form of Equation (Equation 22), where all carbon-containing chemical species are given in carbon moles. This entails the division of all coefficients by 6 (glucose has 6 carbons), except $\alpha_{3}$, which is divided by 3, because ethanol has 2 carbons. With these converted stoichiometric coefficients, we proceed as described above.

For *E. coli*, the data are extracted from Table 2 in [37]. There, biomass formation rate is given in gram biomass per liter and hour. The exchange rates of substrates and products are given in gram substrate per gram biomass per hour. To convert these numbers into stoichiometric coefficients corresponding to Equation (Equation 15), we first convert these numbers from gram to carbon mole. For biomass formation, the values are divided by the mass of one carbon mole biomass (26.11 g C-mol${\;}^{-1}$). Likewise, the exchange rates are multiplied by the mass of one carbon mole biomass and divided by the molecular mass of the respective substrate. Then, the resulting values are converted to stoichiometric coefficients as follows: $\alpha_{2}$ (C-mol biomass per C-mol glucose) results simply from dividing the growth rate by the specific glucose uptake rate (in C-mol C-mol${\;}^{-1}$ h${\;}^{-1}$). Likewise, $\alpha_{3}$ (for acetate) and $\alpha_{4}$ (for CO_2_) are calculated. Finally, the stoichiometric coefficient for O_2_ ($\alpha_{1}$) is obtained by the oxygen balance of Equation (Equation 15), and the coefficient for H_2_O by the hydrogen balance.

## 3. Results

### 3.1. Calculating Elementary Conversion Modes to Characterize Catabolic Pathways

We use elementary conversion modes (ECMs, see [32,33]) to assess the metabolic capabilities of several genome-scale metabolic networks. To illustrate our approach, we begin our analysis with the *E. coli* core network [38], an *E. coli* core metabolism model of reduced complexity, with only 72 metabolites connected by 95 reactions, of which 20 are exchange reactions. To systematically describe all theoretically possible catabolic routes, we use ecmtool [33] to calculate all ECMs which do not produce biomass. The resulting ECMs describe all possible routes and their stoichiometries by which external substances can be interconverted.

We then calculate for each individual ECM the standard Gibbs free energy of reaction, based on the standard energies of formation estimated by eQuilibrator [34]. The standard energies of catabolism $\Delta_{\mathrm{cat}}G^{\circ }$, normalized to one carbon mole of consumed substrate, are displayed in Figure 2. Most of the catabolic pathways have relatively low energy gradients between approximately 20 and 50 kJ C-mol${\;}^{-1}$. To this large group of pathways belong key catabolic routes, such as the fermentation of glucose to lactate or ethanol. Few catabolic routes exhibit Gibbs free energies of reactions with a higher energy gradient than 50 kJ C-mol${\;}^{-1}$ (17 ECMs). The pathways with the largest absolute Gibbs free energy of reaction are the combustion of glucose ($\Delta_{\mathrm{cat}}G^{\circ }\approx$ −488 kJ C-mol${\;}^{-1}$) and the production of formate from glucose ($\Delta_{\mathrm{cat}}G^{\circ }\approx$ −325 kJ C-mol${\;}^{-1}$). In particular, oxygen-using ECMs belong to the group with the largest absolute $\Delta_{\mathrm{cat}}G^{\circ }$ (compare red bars in Figure 2). As shown in Figure 2, the usage of nitrogen does not appear to be an indicator of whether the respective ECM has a high or low energy gradient.

For each catabolic route, we use the metabolic model to calculate the maximal ATP yield. For this, the exchange reactions are constrained to the stoichiometries of the respective ECM, and subsequently the flux through ATP hydrolysis is maximized (see Theory and Methods; Section 2). The resulting maximal ATP yields per carbon mole substrate are indicated by black crosses in Figure 2. While as a tendency high energy gradient pathways also allow for a higher ATP yield, there are a considerable number of ECMs with very low ATP yield (34 ECMs exhibit a maximal ATP yield of less than 0.1 mol ATP per C-mol).

### 3.2. Thermodynamic Efficiency of Catabolic Routes

To investigate whether these general patterns are also conserved in more complete and therefore realistic genome-scale models, we repeat our analysis for the iJR904 metabolic network model [39] of *E. coli* as well as for the iND750 metabolic network model [40] of the yeast *S. cerevisiae*. Figure 3 illustrates the results for *S. cerevisiae*, obtained with the iND750 model. We identify all ECMs using one of the four carbon sources glucose, xylose, α-ketoglutarate, and pyruvate. Also, here, most ECMs yield a low energy gradient, while those with the highest gradient correspond to the full oxidation of glucose and xylose. Full respiration for both sugars releases around 488 kJ C-mol${\;}^{-1}$, and they are, thus, the ECMs with the highest free energy gradient. These two pathways also display the highest ATP yield per carbon mole (approx. 2.92 mol/C-mol), as indicated in Figure 3, left panel. In contrast to this, most other catabolic routes release energy in a range between 40 and 200 kJ C-mol${\;}^{-1}$ with low ATP production. Among these routes are, besides other metabolic modes, fermentation reactions, such as the metabolization of glucose to lactate or ethanol.

The thermodynamic efficiency η of a thermodynamic engine is defined as the fraction of work generated per input energy provided to power the system. Defining the free energy used to drive ATP synthesis as the useful chemical work, we calculate the thermodynamic efficiency (η, Equation (Equation 6)). We assume the typical value of $\Delta_{r}G_{\mathrm{ATPase}}=$ 46.5 kJ
${\mathrm{mol}}^{-1}$ for ATP synthesis [43]. Further, we approximate the Gibbs free energy of catabolism by the corresponding standard Gibbs free energy of reaction, because changes in substrate and product concentrations in the medium are likely to have only a minor effect on the quantity (see Methods for discussion and examples). The determined efficiencies η are depicted in the right panel of Figure 3. Interestingly, the pathways with the highest energy gradient are not the most efficient. For instance, under full respiration of glucose, only 28% of the released free energy is converted to chemical work producing ATP. In contrast, the fermentation of glucose to lactate exhibits an efficiency of 43%. The fermentation of glucose to lactate is one of the most efficient reactions, with higher efficiencies only found in fermentation processes involving ethanol production.

### 3.3. Deriving Anabolic Information from a Metabolic Network

Likewise, anabolic pathways can also be investigated in separation. In the following, we compare predictions from a macrochemical description of anabolism, which ignore details of the intracellular reaction network, with those resulting from employing genome-scale metabolic models. Specifically, we calculate maximal theoretical yields based only on the chemical properties of the nutrients and the biomass, and then we analyze how closely they can be realized by the actual metabolism of *E. coli* and *S. cerevisiae*.

The theoretical limit to how much carbon in the nutrient can be converted to biomass carbon is given by the degree of reduction. If biomass is more reduced than substrate, a fraction of the substrate carbons needs to be oxidized in order to ensure the overall redox balance. Specifically, if $\gamma_{S}$ and $\gamma_{X}$ are the degrees of reduction of substrate and biomass, respectively, then the theoretical maximal yield is given by [10]
(23)$$Y_{X/S}^{\max}=\left\{\begin{array}{cc} \frac{\gamma_{S}}{\gamma_{X}} & \mathrm{for}\;\gamma_{S}\leq \gamma_{X}\\ 1 & \mathrm{else} \end{array}\right..$$

Based on the elemental composition of the biomass and the substrate, an ideal anabolic reaction stoichiometry can be determined. We assume we have a substrate with a sum formula $\begin{array}{c} \left[S\right] \end{array}=\begin{array}{c} {\mathrm{CH}}_{x}O_{y} \end{array}$ (normalized to C-mol) and biomass with $\begin{array}{c} \left[X\right] \end{array}=\begin{array}{c} {\mathrm{CH}}_{a}O_{b}N_{c} \end{array}$, $\gamma_{S}=4+x-2y$ and $\gamma_{X}=4+a-2b-3c$ (assuming ammonia as nitrogen source, see [10]). If $\gamma_{S}\leq \gamma_{X}$, the stoichiometry reads
(24)$$\frac{1}{Y_{X/S}^{\max}}\left[S\right]+b\;{\mathrm{NH}}_{3}⟶\left[X\right]+\left(1+\frac{1}{Y_{X/S}^{\max}}\right){\mathrm{CO}}_{2}+b_{4}\;H_{2}O,$$
with $b_{4}=\left(x/Y_{X/S}^{\max}+3b-a\right)/2$ (see Theory and Methods; Section 2).

This equation allows for the calculation of the standard energy of reaction of anabolism, $\Delta_{\mathrm{ana}}G^{\circ }$. To estimate the Gibbs free energy of formation of biomass, which is required to determine the energy of reaction, we employ the empirical method proposed by Battley [44].

Constraint-based models can be employed to investigate to what extent such an ideal anabolic reaction can be realized by a microorganism’s metabolism. We employ the genome-scale networks for *S. cerevisiae* and *E. coli* and minimize the nutrient uptake for fixed biomass production, while allowing ATP to be provided externally (see Theory and Methods; Section 2). A subsequent optimization, in which the minimal nutrient uptake is fixed and the required reverse ATP hydrolysis is minimized, allows for determining the minimal ATP requirement per carbon mole biomass formed. For the iJR904 model of *E. coli* metabolism, we obtain the following optimal anabolic stoichiometry for growth on glucose: (25)$$\begin{array}{cc} \; & 1.0530\left[S\right]+0.0046O_{2}+0.2577{{\mathrm{NH}}_{4}}^{+}+0.0220{{\mathrm{HPO}}_{4}}^{2-}+0.0056{{\mathrm{SO}}_{4}}^{2-}\\ \; & \;⇌{\left[X\right]}_{\mathrm{eco}}+0.0530{\mathrm{CO}}_{2}+0.5635H_{2}O+0.2205H^{+}, \end{array}$$
where [S] denotes 1 C-mol of substrate ($\frac{1}{6}\begin{array}{c} C_{6}H_{12}O_{6} \end{array}$) and [X]_eco_ 1 C-mol of *E. coli* biomass, with the sum formula determined by the biomass reaction of the iJR904 model
(26)$${\left[X\right]}_{\mathrm{eco}}=CH_{1.811}O_{0.503}N_{0.258}P_{0.022}S_{0.006}^{0.018-},$$
and a degree of reduction and energy of formation of
(27)$$\gamma_{X,\mathrm{eco}}=4.193,\;\Delta_{f}G_{X,\mathrm{eco}}^{\circ }=-101.10\;\mathrm{kJ}\;{C-\mathrm{mol}}^{-1}.$$

The resulting simulated maximal yield of $Y_{X/S}^{\mathrm{sim}}=\frac{1}{1.053}=95.0\%$ is slightly lower than that expected by Equation (Equation 23), which results in $Y_{X/S}^{\max}=\frac{4}{4.193}=95.4\%$. This is explained by the fact that small amounts of oxygen are also required for pure anabolic biomass formation. In iJR904, this is caused by a minimal required flux through a cytochrome oxidase, which requires molecular oxygen as substrate.

A subsequent optimization reveals a minimal requirement of 1.766 mol ATP per carbon biomass produced.

For the iND750 metabolic model of *S. cerevisiae*, we obtain
(28)$$\begin{array}{cc} \; & 1.0718\left[S\right]+0.0518O_{2}+0.1557{{\mathrm{NH}}_{4}}^{+}+0.0055{{\mathrm{HPO}}_{4}}^{2-}+0.0022{{\mathrm{SO}}_{4}}^{2-}\\ \; & \;⇌1{\left[X\right]}_{\mathrm{sce}}+0.0718{\mathrm{CO}}_{2}+0.4035H_{2}O+0.1404H^{+}, \end{array}$$
with
(29)$${\left[X\right]}_{\mathrm{sce}}=CH_{1.8243}O_{0.6589}N_{0.1557}P_{0.0055}S_{0.0022}$$
and a degree of reduction and energy of formation of
(30)$$\gamma_{X,\mathrm{sce}}=4.080,\;\Delta_{f}G_{X,\mathrm{sce}}^{\circ }=-128.76\;\mathrm{kJ}\;C-\mathrm{mol}{\;}^{-1}.$$

Here, the discrepancy between the computationally determined maximal yield of $Y_{X/S}^{\mathrm{sim}}=\frac{1}{1.0718}=93.3\%$ and the $Y_{X/S}^{\max}=98.0\%$ expected from Equation (Equation 23) is even larger. The minimal requirement of ATP to produce biomass is predicted to be slightly larger than for *E. coli* with 2.031 mol ATP per C-mol biomass.

We repeated the calculations for different carbon sources. The results are summarized in Table 1. In general, the expected trend can be observed that more oxidized carbon sources result in lower maximal yields (see also Equation (Equation 23)). Moreover, the maximal simulated yields predicted by the model are usually very close to the maximal yield predicted by the degree of reduction alone. This demonstrates that the metabolic networks of *S. cerevisiae* and *E. coli* are structured in such a way that the theoretical yield limits, resulting from the degrees of reduction of substrate and biomass alone, can be closely approximated. Only for *S. cerevisiae* growing on oxoglutarate, yields predicted by the model are considerably lower. The reason for this is that the network defined by the iND750 model is not capable of producing biomass from oxoglutarate without metabolic side products. The optimal solution produces 0.073 mol xanthine (C_5_H_4_N_4_O_2_) per C-mol biomass. This leads to a reduced carbon (and in fact, nitrogen) yield, but a larger free energy gradient of anabolism compared to other carbon sources, which can be readily explained by the highly oxidized side product.

### 3.4. Separating Catabolism from Anabolism Based on Chemostat Data

In a controlled continuous microbial cultivation system, such as a chemostat [45,46], it is possible to grow microbial cultures at a steady state with pre-defined growth rates. In chemostat cultures, the dilution rate *D* is an experimental control parameter and, when the culture has reached a stationary state, the specific growth rate is forced to be equal to the dilution rate. Measuring nutrient and gas exchange rates as well as nutrient and product concentrations in the reactor allows for the experimental determination of the overall growth stoichiometries [11,12,13,19].

In the following, we employ experimentally determined macrochemical equations for the growth of *S. cerevisiae* [36] and *E. coli* [37] to calculate catabolic stoichiometries, ATP production potential, and thermodynamic efficiencies for each condition. Both publications contain a full characterization of macrochemical growth parameters for chemostat cultures at different dilution rates. Specifically, gas exchange rates were quantified to determine oxygen uptake and CO_2_ evolution rates, and substrate, product, and biomass concentrations in the medium were measured, allowing for the calculation of the respective exchange rates. Together with the elemental composition of the biomass, which is also reported in both sources, these data provide a precise quantification of the macrochemical growth equation for different dilution rates. We then separated the macrochemical equations by first identifying the ideal anabolic stoichiometry based on the degrees of reduction of substrate and biomass (see Methods, Section 2.3) and then subtracting this anabolic stoichiometry from the macrochemical equation (see Methods, Section 2.5).

The determined catabolic coefficients are summarized in Figure 4. It can clearly be seen that the onset of overflow metabolism, when glucose is partly fermented even in the presence of sufficient oxygen, occurs at growth rates of around 0.3 $h^{-1}$ for *S. cerevisiae* and around 0.4 $h^{-1}$ for *E. coli*. With the catabolic coefficients, we calculate the standard Gibbs free energy of reaction of the overall catabolic conversion, where we obtained the standard Gibbs free energies of formation, required for this calculation, from the equilibrator tool [34]. It can be observed (blue lines in Figure 4) that with the onset of overflow metabolism, the Gibbs free energy gradients are also reduced significantly.

### 3.5. Is the Linear Energy Converter a Good Model for Microbial Growth?

With the theory developed on how to separate catabolism and anabolism from a macrochemical equation, we are now in the position to systematically test whether experimental data of chemostat growth are actually in agreement with the assumption of a linear flux–force relationship (see Equation (Equation 1)) between the anabolic and catabolic driving forces and the corresponding fluxes.

For this, we will focus on the experimental data for *S. cerevisiae* [36]. The reason for this is that in the data for *E. coli* [37] the carbon recovery coefficient, which quantifies the fraction of carbons ‘recovered’ in the biomass and metabolic byproducts, compared to the carbons provided by the nutrients, was lower than 90% for large growth rates (above ≈$0.4\;h^{-1}$). This indicates that not all metabolic products were measured, thus preventing a reliable calculation of catabolic stoichiometries.

In the thermodynamic interpretation of microbial growth as a linear energy converter [18,19], catabolic and anabolic fluxes are assumed to linearly depend on the catabolic and anabolic forces, as illustrated by Equation (Equation 1). Here, the anabolic flux equals the growth rate, which in turn is set by the dilution rate of the chemostat, $J_{\mathrm{ana}}=D$. The catabolic flux is calculated by separating the catabolic and anabolic equations (see above). As discussed in Section 2 (Theory and Methods), we approximate the actual Gibbs free energies by the standard energies, and we motivate this approximation by the large overall energy gradients. Moreover, we assume that the anabolic Gibbs free energy $\Delta_{\mathrm{ana}}G$ is approximately constant over different dilution rates, because the stoichiometry of anabolism remains constant, according to Equation (Equation 8). Based on the experimentally determined biomass composition of *S. cerevisiae* [36] (CH_1.79_O_0.57_N_0.15_), we calculate the standard energy of formation of biomass with the empirical method of Battley [44] as
(31)$$\Delta_{f}G_{X}^{\circ }=-104.9\;\mathrm{kJ}\;{C-\mathrm{mol}}^{-1},$$
and with that the anabolic energy of reaction as
(32)$$\Delta_{\mathrm{ana}}G^{\circ }=-42.2\;\mathrm{kJ}\;{C-\mathrm{mol}}^{-1}.$$

With this full knowledge of catabolic and anabolic fluxes and forces, we can challenge the linear converter model. In Figure 5 (Appendix A for *E. coli*), various fluxes (catabolic glucose consumption, anabolic (growth) rate, total glucose consumption) are displayed as functions over the catabolic driving force or, alternatively (*x*-axis on top), the force ratio $x=\Delta_{\mathrm{cat}}G^{\circ }/\Delta_{\mathrm{ana}}G^{\circ }$. It can clearly be observed that the fluxes do not linearly depend on the forces, as would be predicted by the linear converter model. On the contrary, fluxes are larger for smaller forces. This indicates that the lower the energy gradient driving the growth, the faster microbes grow under the condition of overflow metabolism. This stark discrepancy from a linear converter model can readily be explained by considering that overflow metabolism results from active regulatory processes inside the cells. Often, overflow metabolism is explained by capacity constraints within the cell: whereas respiration results in a considerably higher yield, it also requires higher protein investment than fermentation, and therefore, at very high growth rates, it is more efficient to ’waste’ substrate and operate a lower yield pathway (see, e.g., [2,47,48]). This entails that there exist feedback regulation mechanisms which result in a highly non-linear relation between the observed fluxes and the driving forces. It can be concluded that a linear energy converter model is too simplistic and is not in agreement with experimental data, at least over conditions in which the catabolic pathways change. The reason for this is to be sought in feedback mechanisms by which cells adapt their metabolism to external conditions.

Similar to the calculation of the maximal ATP yields of the different catabolic pathways, we determine the maximal ATP yield for the observed catabolic stoichiometries by constraining the genome-scale networks to the observed catabolic stoichiometries. The red circles in Figure 3 present the result of this calculation (left panel) and the respective thermodynamic efficiencies (right panel). Interestingly, the ATP yields as well as the efficiencies are higher than those for elementary conversion modes with similar energy gradients. This can be explained by considering that the experimentally observed catabolic stoichiometries are a linear combination of two conversion modes (full respiration and fermentation) only and that the fermentation pathways are especially identified to have the highest thermodynamic efficiency (see Figure 2 and Figure 3).

With the usual definition of the power as the product of flux and force, we can quantify the catabolic and anabolic powers $P_{\mathrm{cat}}=-J_{\mathrm{cat}}\;\cdot \;\Delta_{\mathrm{cat}}G$ and $P_{\mathrm{ana}}=-D\;\cdot \;\Delta_{\mathrm{ana}}G$, as well as the power of ATP production $P_{\mathrm{ATP}}=c_{\mathrm{ATP}}\;\cdot \;J_{\mathrm{cat}}\;\cdot \;\Delta_{r}G_{\mathrm{ATPase}}$ in kJ C-mol${\;}^{-1}$. Figure 6 shows that the powers increase approximately linearly with growth rate, despite the fact that metabolism changes considerably for fast growth rates. Moreover, *E. coli* and *S. cerevisiae* behave quite similarly. For large growth rates, the powers of *E. coli* appear to be somewhat lower, but this could be a result of incomplete carbon recovery in the experiments, because besides acetate no other fermentation products were measured. This could lead to an underestimation of the Gibbs free energy gradients and consequently of the ATP yields.

## 4. Discussion and Conclusions

Microbial organisms are a cornerstone of the modern biotechnological industry. They are invaluable for producing pharmaceuticals, food, and construction materials [3]. Today, by using sophisticated genetic techniques and engineering, microorganisms can be used to tackle modern problems of society, such as the remediation of waste land, the production of drugs, or finding environmentally sustainable building materials [49,50,51]. However, such advancements in exploiting bacteria and unicellular eukaryotes were only possible with a thorough understanding of their metabolism. Multiple theories and techniques have been developed to gain knowledge about metabolic pathways. One of the most promising strategies currently is to develop genome-scale metabolic networks encoding almost all metabolic information available for an organism [52].

Genome-scale metabolic models are used to understand and probe the metabolic capabilities of an organism and allow for the calculation of maximal yields [53]. The construction of such models only became possible with advances in sequencing technologies and protein function prediction, through which more and more fully sequenced genomes and better annotated proteomes become available [20,54]. Before genome-scale metabolic networks existed, many scientists and engineers relied on macrochemical equations. These equations describe the metabolism of organisms as a black box by one overall chemical equation. Over the decades, this approach has been characterized extensively for its applicability and in the context of thermodynamics. However, a challenge when applying macrochemical equations is separating anabolism from catabolism so that both metabolic modes can be studied individually. Here, with the help of genome-scale models and thermodynamic calculations, we showed how we can extract both anabolic and catabolic conversions. We characterized both metabolic modes and challenged commonly used viewpoints on microbial metabolism, such as its representation as a linear energy converter.

By using elementary conversion modes (ECMs), which are an alternative to elementary flux modes, we could systematically enumerate catabolic routes from genome-scale networks of *E. coli* and *S. cerevisiae* (see Appendix A and Figure 3). Combined with thermodynamic data of the Gibbs free energy of formation for all metabolites, as provided by the eQuilibrator tool, it is simple to derive the standard Gibbs free energy of reactions for all input–output relationships (ECMs). By doing so, we calculated the catabolic driving force of microbial growth for all theoretically possible routes.

However, the second law of thermodynamics implies that not all of the available free energy can be used to perform useful chemical work. Combining ECMs with constraint-based modeling of genome-scale networks, we calculated the thermodynamic efficiency of the ATP production of each catabolic route. Interestingly, we found the most efficient pathways exhibited a thermodynamic efficiency between approximately 30 and 45%. Interestingly, most catabolic routes show a considerably lower efficiency below 20%. It should be noted, though, that the values for the efficiency have to be interpreted with care. Firstly, we assumed standard energies of reaction for the catabolic routes. The actual concentrations of nutrients and catabolic products in the medium may slightly affect these energy values. Secondly, we assumed a fixed value for the energy of reaction for ATP synthesis, which of course may change for different physiological conditions and depends primarily on the ATP–ADP ratio and the concentration of inorganic phosphate. Taking this into account, the highest efficiency, which is observed for the fermentation pathways, is very close to the 50% that is predicted to yield the highest ATP production rates by simple linear thermodynamic energy converter models [55]. It is remarkable that the actually realized catabolic pathways in chemostat cultures (see red circles in Figure 3) provide a higher efficiency than elementary pathways with a similar free energy gradient. This observation stresses the important role of the pure respiration and fermentation pathways of catabolism. Because of their high efficiencies, operating them in combination always provides a higher thermodynamic efficiency than any single elementary conversion mode.

While a linear energy converter model seems adequate to predict optimal thermodynamic efficiencies of ATP-producing pathways with reasonable accuracy [55], our interpretation of experimental data shows that this is not the case when microbial growth is considered. Our results clearly demonstrate that the flux–force relationship is not linear and that, in fact, anabolic and catabolic fluxes *decrease* with increasing catabolic driving force. In other words, the faster microbes grow, the lower the energy gradient that drives this growth. This observation, however, holds for conditions during which catabolism exhibits rather drastic changes, from pure respiration at low growth rates to predominant fermentation at high growth rates. For growth in batch cultures on different substrate concentrations, however, it was shown that the linear converter model yielded very good results, which indeed fit the data better than a simple Monod equation [56]. It can therefore be hypothesized that the linear energy converter model is adequate as long as the catabolic mode does not change, and thus the driving force is mainly influenced by substrate concentration, but fails as too simplistic if experimental conditions encompass a change in catabolic pathways. In order to experimentally test this hypothesis, systematic growth data for different microorganisms grown under different nutrient concentrations and dilution rates would be necessary. As discussed in detail in the Results section, such experiments must be designed to yield exact information about the media composition, growth rate, biomass composition, and formation rates of metabolic product. In short, all required information to obtain a precise macrochemical equation of metabolism is required to calculate the anabolic and catabolic fluxes separately, which allows for testing the validity of the linear converter model.

An interesting observation is that the output powers scale approximately linear with growth rate and that the proportionality is very similar for two organisms as different as the bacterium *E. coli* and the eukaryote *S. cerevisiae*. For the anabolic power (growth rate times anabolic driving force), this result is trivial because we assumed the anabolic force, $-\Delta_{\mathrm{ana}}G$, to be constant. However, for the catabolic power (nutrient consumption rate for catabolism times catabolic driving force), this result is far from obvious. For technical systems, such as ships [57], bikes, cars, or trains (see [58], Chapter IIIA), the power increases over-proportionally with speed, approaching an approximately quadratic relationship. The linear power–growth rate relationship entails that, employing engineering terms, a ‘resistance’ that needs to be overcome by the thermodynamic driving forces when producing new biomass is a constant rather than dependent on the biomass production rate. Moreover, the force, corresponding to the slope of the power–growth rate curves, appears to be the same for *E. coli* and *S. cerevisiae*. Whether these laws are of a universal nature remains to be tested with systematic experiments of more microbial species grown on different nutrient sources.

The present study is largely theoretical and it provides advances in our methodology of how to study microbial metabolism, both through the more classical black box approach based on macrochemical equations as well as with constraint-based models. A direct application of our theories helps in predicting maximal yields for different nutrients and thus may also support the design of cost- and resource-efficient growth media. A recent approach to avoid the emergence of non-producing mutants in biotechnological applications is to genetically engineer microbes such that product formation is directly coupled with growth [59,60,61]. Our theories are directly applicable to such scenarios and allow for estimating the expected effects on yield and thermodynamic driving forces.

From a theoretical perspective, a main challenge in the future will be to understand the underlying mechanisms that cause microbial growth to deviate so strongly from the behavior expected for a linear energy converter. The next challenge will then be to employ such understanding to find equations that generalize the simplistic linear converter model to also include scenarios in which catabolic pathways change. Such equations would be highly useful, because they will allow for predicting not only the yield but also the growth rates of microorganisms.

In summary, we could show how combining black box macrochemical approaches and genome-scale metabolic models can help to systematically characterize catabolic routes and find separate chemical equations for anabolism and catabolism. Interpreting experimental data from chemostats with our theoretical models reveals that the efficiency of catabolism appears optimal, both for *E. coli* and the yeast *S. cerevisiae*, over a wide range of growth rates. Moreover, our analyses allow us to speculate that the linear power–growth rate relationship is a universal property of microbial growth.

## Figures and Tables

**Figure 1 life-14-00247-f001:**
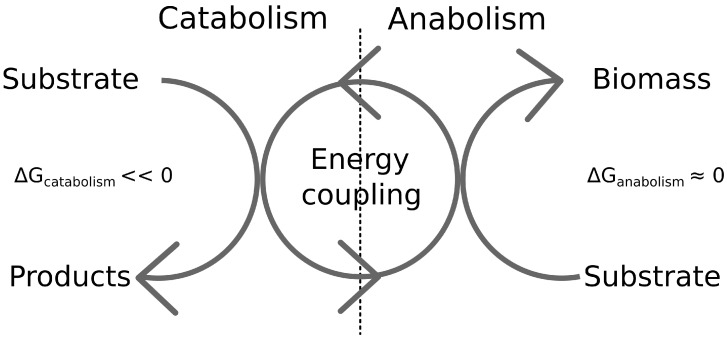
The view of microbial metabolism as a thermodynamic energy converter. Catabolic reactions have a large negative free energy gradient, driving anabolic reactions [14].

**Figure 2 life-14-00247-f002:**
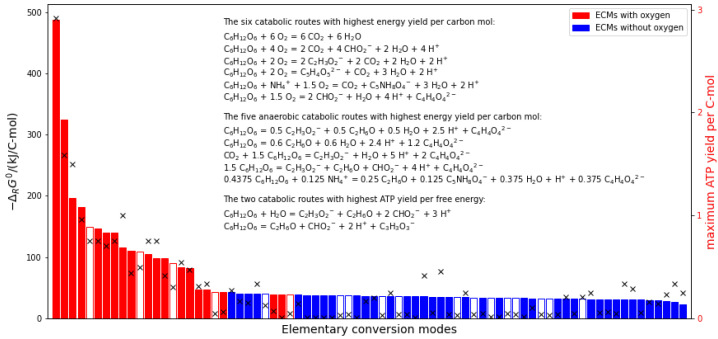
Standard Gibbs free energies of all catabolic pathways, normalized to carbon mole. The catabolic pathways were derived using elementary conversion modes (ECMs) calculated from the *E. coli* core network. Red symbolizes ECMs that use oxygen, while blue denotes ECMs not using oxygen. Filled bars belong to ECMs that include no compounds with the element nitrogen, while empty ones include nitrogen-containing metabolites. The black crosses indicate the maximal yield of ATP per carbon mole nutrient for each ECM (right axis).

**Figure 3 life-14-00247-f003:**
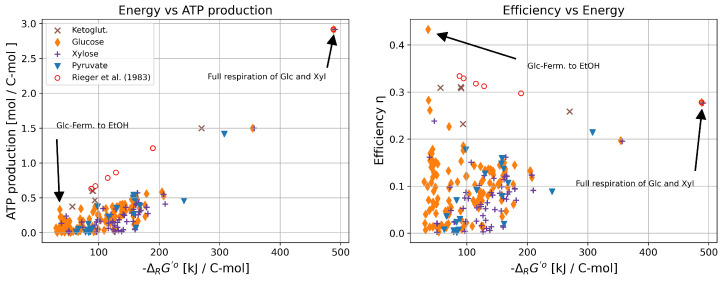
Thermodynamic characterization of catabolic routes in the *S. cerevisiae* genome-scale model (iND750) for α-ketoglutarate, glucose, xylose, and pyruvate as carbon sources. Additionally, oxygen is allowed to be a substrate in the calculation of the elementary conversion modes. The efficiency is based on a typical value of 46.5 kJ/mol for the production of ATP in *E. coli* [43]. For comparison we also calculated ATP production and efficiency based on data by [36]. Only ECMs are plotted with an ATP yield higher than 0.001 mol/C-mol.

**Figure 4 life-14-00247-f004:**
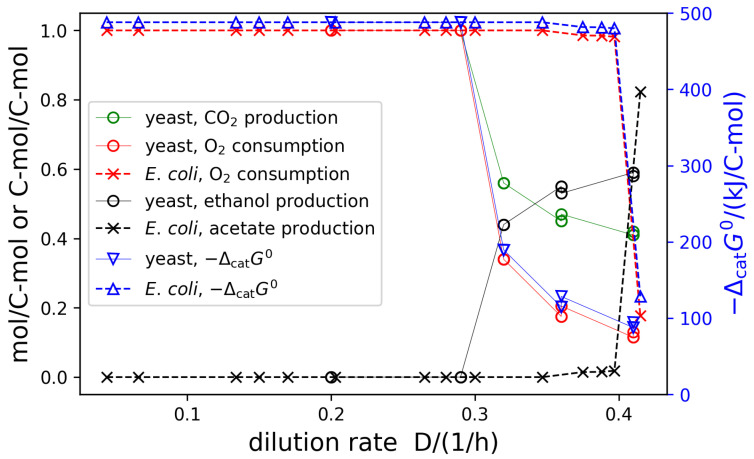
The catabolic stoichiometric coefficients and thermodynamic driving forces determined for the chemostat growth of *E. coli* [36] and *S. cerevisiae* [37] at different dilution rates. All coefficients are given in mol/C-mol substrate, except for ethanol and acetate, which are given in C-mol/C-mol substrate. For *E. coli*, CO_2_ production is identical to O_2_ consumption. The thermodynamic driving force is given as the standard energy of reaction of the overall catabolic conversion, normalized to one carbon mole of substrate.

**Figure 5 life-14-00247-f005:**
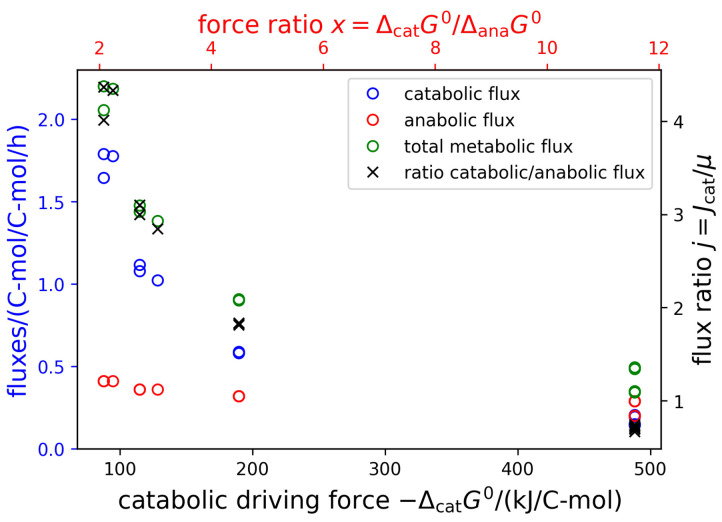
Metabolic fluxes as functions of the catabolic driving force. Shown are the catabolic (blue), anabolic (red), and total (green) glucose consumption rates in dependence of the catabolic driving force, $-\Delta_{\mathrm{cat}}G^{\circ }$. On the *x*-axis on the top, the force ratio $x=\Delta_{\mathrm{cat}}G^{\circ }/\Delta_{\mathrm{ana}}G^{\circ }$ is given.

**Figure 6 life-14-00247-f006:**
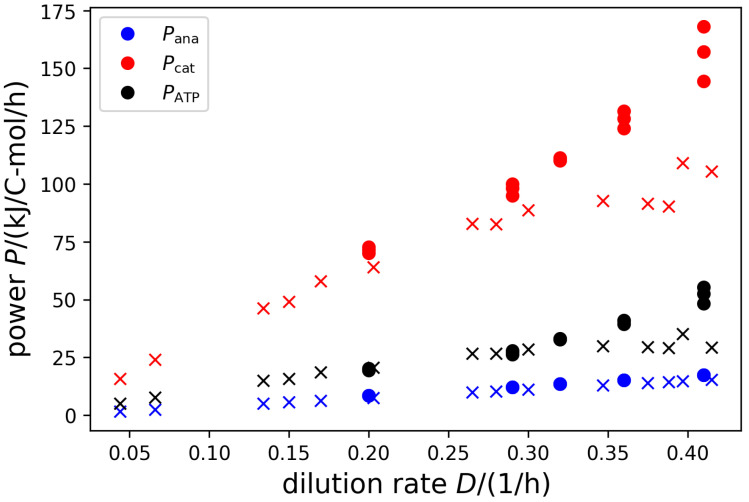
Catabolic and anabolic powers, as well as the power of ATP synthesis as a function of growth rate. Catabolic power is depicted in red, anabolic power in blue, and ATP synthase power in black. Circles present results for *S. cerevisiae*, crosses for *E. coli*.

**Table 1 life-14-00247-t001:** Thermodynamic properties of anabolic pathways. The theoretical maximal yield $Y_{X/S}^{\max}$ is calculated according to Equation (Equation 23). The maximal yield $Y_{X/S}^{\mathrm{sim}}$ predicted by the metabolic model is determined using the linear program (Equation 12). The minimal anabolic ATP requirement per carbon mole biomass $a_{\mathrm{ATP},\min}$ is determined using the linear program (Equation 13). The standard energies of reaction for anabolism, $\Delta_{\mathrm{ana}}G^{\circ }$, are determined from the overall anabolic stoichiometries like those given for glucose in Equations (Equation 25) and (Equation 28).

Organism	Carbon Source	$Y_{X/S}^{\max}$	$Y_{X/S}^{\mathrm{sim}}$	$a_{\mathrm{ATP},\min}$ (mol C–mol${\;}^{-1}$)	$\Delta_{\mathrm{ana}}G^{\circ }$ (kJ C–mol${\;}^{-1}$)
*E. coli*	glucose	0.954	0.950	1.766	−54.81
	xylose	0.954	0.950	1.813	−56.74
	oxoglutarate	0.763	0.730	2.051	−44.28
	pyruvate	0.795	0.791	2.268	−32.44
*S. cerevisiae*	glucose	0.980	0.933	2.031	−74.51
	xylose	0.980	0.933	2.245	−77.85
	oxoglutarate	0.784	0.457	1.453	−279.33
	pyruvate	0.817	0.815	2.684	−29.62

## Data Availability

The original contributions presented in the study are included in the article/Appendix A; further inquiries can be directed to the corresponding authors. The code can be found at https://gitlab.com/qtb-hhu/thermodynamics-task-force/2023-energy-metabolism-of-microorganisms, last accessed on 4 January 2024.

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
