# Peer review of "Microbial Pathway Thermodynamics: Stoichiometric Models Unveil Anabolic and Catabolic Processes"

_life, 2024, doi:10.3390/life14020247_

Round 1

Reviewer 1 Report

Comments and Suggestions for Authors

As an experimental scientist on microbial metabolites, I have difficulty in reviewing the theoretical content of this manuscript. Therefore, I paid more attention on the validation and potential applications of the mathematic modeling from this study, and I was not very satisfied on what I read in this manuscript.

1. The authors relied on the experimental data from two publications (references 29 and 30) on the metabolic flux of glucose fermentation in yeast and E. coli, respectively, to validate the proposed models. However, the contents and data structures of these two studies were barely introduced and explained in this paper. Therefore, the sections 2.9 and 2.10 were hard to follow and understand. A better work can be done on this issue.

2. Current discussion mainly cover the theoretic value of proposed mathematic modeling, but did not provide a good idea on its practical value as well as explain what are the experimental data needed for its application. Please expand the discussion on this issue further.

A minor issue. Blue and red colors are used in Figure 1. Please indicate the meaning of these two colors in the figure legend.

Comments on the Quality of English Language

The quality is fine.

Author Response

Please see the attachment. An additional file in which the differences are marked by color was  submitted as material not intended for publication.

Reviewer 2 Report

Comments and Suggestions for Authors

The manuscript "Microbial pathway thermodynamics: structural models unveil anabolic and catabolic processes" describes strategies to use genome-scale metabolic ECM models to analyze anabolism and catabolism processes seperately as an improvement of the previous macrochemical equations which treats microbial metabolism as a black box. By doing so, the authors determined thermodynamic efficiencies and evaluated approximate biomass yield, as well as uncovered the limitations of the linear energy converter model. This work appears to be a good contribution to the field and this reviewer just have a few comments:

1. In the title, the author used the word "structural models", however, this reviewer, at the beginning, got confused this with three dimensional structural models as commonly used for biomolecules. Perhaps "stoichiometric metabolic models" is more appropriate, or at least this should be structural metabolic models.

2. Please give a more detailed description of "ECM" other than just citing the reference in the introduction which will be beneficial for broader readers.

3.  One important discovery the author made using their new models is to challenge the "linear energy converter" model (line 49-50). Again, the authors cited multiple papers for this model. It seems pertinent to give a more detailed description about this linear energy converter. What does "these generalized forces and fluxes is often assumed to be linear" mean?

4. As the author pointed out in the introduction and discussion sections, a better understanding of the microbial metabolism processes is critical for industrial usage of these microbial systems. What are the potential contribution(s) of the new understandings from the paper to that point? It will be more compelling if the authors could add this to the discussion.

Author Response

(The authors gave the same response as above.)

Reviewer 3 Report

Comments and Suggestions for Authors

The work performed by Ebenhöh et al. deals with the modelling of core metabolic routes of two model organisms, using an approch that is intermediate with respect to macrochemical equations and genome-scale metabolic models. In short, the authors employed essential conversion modes (ECMs), a recent way to represent metabolism by using multiple macrochemical-like representations (each with its stochiometry, DeltaG, and normalized by carbon-mole) of different metabolic modes. Each metabolic mode is different for input substrate and ouput products of metabolism. These ECMs are a coarse-grained representation that "aggregates" all the single catalyzed reactions and solute transportations that connect external substrate and products in genome-scale metabolic models. This avoids the problem of genome-scale metabolic models that need to integrate over all fluxes up to reach a steady state, as this operation is computationally "exploding" in complexity when the n° of interconnected reactions of the metabolism increases, becoming analitically intractable for full interconnected metabolic networks of organisms. Despite being technical interesting, there are some parts that are not completely clear and the use of ECMs intead of fluxes does not permit to explicitly treat the concentration of metabolites: so, as the authors relied on standard conditions, which are indeed not biological, they have to repeat the calculations (at least for one of the organism, such as the core model of E. coli) using two different input/ouput concentrations (ideally when [inputs] are significantly higher than [products], or [products] are significantly higher than [substrates]), so that their model can be validated to be robust over the non-biological assumptions they made. If the model can be made robust to assumptions violations, the conclusions are interesting for accounting on specific cases in which linear macrochemical metabolic models may result not useful to interpret and optimize the metabolism in microbial fermentations, especially for organisms other than E. coli or S. cerevisiae.

The full list of points:

- In general: please try to improve the introduction section: I am one potentially interested reader of your work, but, if not involved in the reviewing process, reading the first ten lines could have made me to skip reading your work, regardless of your interesting results

- Line 23 "Not surprising, therefore, that much scientific effort has been spent to understand the metabolism of microbes necessary for the chemical interconversion of substances" -> PLEASE IMPROVE THE SYNTAX. "Therefore, it is not surprising that such ...."
AND REMOVE "necessary for the chemical interconversion of substances". IT IS REDUNDANT AND NOT NECESSARY FOR THE READER OF THIS JOURNAL

- Lines 25-26: "Today, the human exploitation of microbial metabolism has long left the stage of merely producing fermented products" Not clear, as "fermentation" can be used in differnet contexts with different meanings. Please be more clear and provide a ref.

- Line 30: please explain in the introduction what you intend for the dependency over "metabolic capabilities" (e.g., efficiency, plasticity, specificity, ...). Thermodynamic constraints cannot be considered a controllable dependence, as they are embedded in these capabilities: during microbial evolution the metabolism has been shaped by those constraints as an embedded component of the metabolic capabilities (i.e., I can increase fitness by selecting on enzymes or proteins that allow a different thermodynamic efficiency or plasticity of the same macro-route in certain conditions, but not by selecting for different thermodynamic impositions, which is a constant).

- Line 32-33: explain why it is a complex scientific problem or provide a ref.

- Line 41: "biotechnologically important" so biotechnologically unimportant metabolic properties cannot be understood?

- Figure 1: "Catabolic reactions have a large negative free energy gradient, driving anabolic reactions" This is actually not understandable from the figure

- Line 44: "the formation of new biomass from the nutrients" Is also ATP used? Please add it.

- Line 51 "Onsager has shown that the linearity holds in general for systems close to equilibrium". So, what is Onsager saying for non-equilibrium systems, as cited in the previous statement? I think is more important to cite any theory on non-equilibrium systems

- Line 53: Considering the order of sentences and the syntax, It is difficut to catch what "this approximation" refers to. Please rewrite this part better and place the flux of arguments in the right order

- Line 56: "principle thermodynamic limitations" ??
Not clear what is intended for "principle". "Initial"?, "Essential"?, "Mandatory constraint"?. Please be more explicit

- Line 57: "Describing macrochemical, catabolic and anabolic equations experimentally"
Not clear

- Line 64: "This information greatly facilitates building". Well, I would say "This information allows building...".
How could someone build a metabolic model with a non-culturable organism?

- Line 65: "a stoichiometric" -> "stoichiometric"

- Line 66-67: "optimal flux distributions optimizing some objective function" Bad syntax

- Theory and Methods: 2.1 Calculating elementary conversion modes:
The authors considered only the core reactions for the use and production of carbon-based compounds, excluding any ECMs that involves an external substrate or product compounds that contain phosphate, sulphur, nitrogen, and any complex Carbon-based compunds (e.g., oligo-/poly- saccharides, also from internal structural or reservoir components). This is leading to a major semplification over the microorganism metabolism, leading to more negative catabolic Deltas than what is expected from experimental biomass measurements. So the authors have to provide such limitation to essential/core reactions in the objectives of their work and discuss also the implications for this simplification. Some assumptions are reported, bu this is not clearly stated.

- 2.2: how are reactant and product concentrations treated? The DeltaG0 free energies are usually at 1M concentration of the reactant and product, which are far from the real, non-equilibrium, concentration found in the organism metabolism

- Equation (4) -> cATP is not defined here, but in the results

- Lines 178-183: this should be placed in the introduction or discussion section

- FIgure 2: please use consistent definitions to improve on readability. (e.g., is "catabolic route" used to intend an "elementary conversion mode"; is ATP yield per free energy the same as "thermodynamic efficiency of ATP production"?)
Are these models full deterministic or could you introduce some confidence estimate? For instance, by varying the concentrations of reactant and products with respect to standard conditions.

- Line 197: "around 17 ECMs". Why "around", you should exaclty know how many of them

- Equation 20 is redundant with respect to methods section

- Lines 232-233: "we approximate the Gibbs free energy of catabolism by the corresponding standard Gibbs free energy of reaction, because changes in substrate and product concentrations in the medium are likely to have only a minor effect on the quantity" this part should be reported in the methods section. Anyway, I disagree. How could you make such an approximation. There are metabolic reactions at molecular level that are esoergonic only in physiological conditions. I think Equilibrator will allow to calculate DeltaG values in conditions different from standard. You have to show the effect (on ECMs DeltaG) of using different concentrations as constraints. Your model does not explicitly consider concentrations and assumes them to be in non-biological state. So, you need to show the model is robust to the choice of markedly different concentrations schemes

- Lines 281-282: "In general, the expected trend can be observed that the more oxidised carbon sources result in lower maximal yields"
Considering that the final biomass is the same (so it has the same reduction level), this is an obvious consequence from (21), without any modelling of anabolism

- Lines 283: "Moreover, the maximal yields predicted by the model are usually very close to the maximal yield predicted by the degree of reduction
alone." Is this indicating that your model is a good approximation? Please be more specific on the comments, as the reader cannot understand the relevance of your results

- Line 288: "a larger free energy gradient" This could be due to using a model in which a consitent amount of energy is considered useful only for biomass production, excluding reservoir polymers and secondary metabolites

- Please provide a Methods section for 2.10 and simplify the relative paragraph. The actual structure makes very difficult to follow what you are presenting in this part of the manuscript, as in the previous paragraph you theoretically calcualte the Deltas per different ECMs, while here you are presenting a macrochemical calculation that can be fitted on DeltaG and experimental data of biomass formation (and given substrate concentration?). It is not clear how the previous calcualted Deltas of catabolic and anabolic processes are connected to this paragraph

- Lines 310-312: "In the original publication that we draw our data from for E. coli [30], also higher dilution rates were investigated"
Not clear, please explain improve the explanation and specifiy dilution rate of what

- Line 314: what do you mean for "carbon recovery"?

- Line 322: "phenomenolocigal" ?

- Line 322 "Considering the large energy gradients, we approximate the actual Gibbs free energies by the standard energies" What is the energy gradient?

- Lines 324-325: so you assume that cells are in exponential growth and will keep the same rate under different dilutions. Which are the corresponding OD600 to which these different dilutions correspond to?

- Line 335 "On the contrary, fluxes are larger for smaller forces" There seems to be a linear relation, but this is inverse to what expected. (You actually commented on this in the discussion section, but why not include a comment also here?)

- "This entails that there exist feedback regulation mechanisms, which are highly non-linear."
 I think that they can be linear, as your model would not have been able to account for these regulations (i.e., sulphur, nitrogen or reservoir carbon production is not accounted and may represent accessory non-core metabolic routes that are differently activated in different conditions)

- Line 345: "The reason for this is to be sought in non-linear feedback mechanisms by which cells adapt their metabolism to external conditions" these feedback change could be linear or not, this cannot be concluded from your models. What you can say is that your model cannot consider some processes (involving small molecule non-carbon-based or complex carbon-based polymers) and these are significantly impacting the DeltaG so that it is not possible to relate it to experimental fluxes. But if you could account for all non-carbon and complex carbon-based biomass, maybe a linear relation can be retrieved in certain conditions.

- From line 367 to 389 the text is perfect for part fo the introduction section and much more clear and appealing than the corresponding part in the actual introduction

- Lines 378-380:"construction of such models only became possible with the advances in sequencing technologies, through which more and more fully sequenced genomes become available"
I would say that the main limitation nowaday is being able to correctly assign a metabolic function to proteins (regulators) and enzymes starting from primary sequences. A sequence is just a sequence, do not encode by itself any metabolic function

- Lines 393-395: "Combined with thermodynamic data of the Gibbs free energy of formation for all metabolites, as provided by the eQuilibrator tool, it is simple to derive standard Gibbs free energy of reactions for all input-output relationships (ECMs)"
Yes, but in standard, non biological, conditions

- Line 399: "Combining ECMs with constraint-based modelling of genome-scale networks"
I would not say that the authors combined the two, they did not perform contraint-based modelling of genome-scale networks, but rather they used the core genome-scale networks as guide to define a model made of coarse-grained relations (ECMs) that permit to theretically reproduce the metabolic network of core transformations, simplifying on integrating the many multiple and interconnected steps of genome-scale metabolic models.

- Lines 405-407: "For one, we assumed standard energies of reaction for the catabolic routes and the 405
actual concentrations of nutrients and catabolic products in the medium may slightly affect 406
these values" This has to be declared in advance of the study, as it is not expected to influence the results only slightly. Moreover, please use at least two different extreme values for the concentrations (the substrate significanyl higher or lower than the product) to repeat the calculation and quantify the actual difference. If conclusions are robust, then the authors can leave the discussion about a slight effect, otherwise they have to change the conclusion and discussion of their results

- Lines 449-451:"Interpreting experimental data from chemostats with our theoretical models reveals that the efficiency of catabolism appears optimal, both for E. coli and the yeast S. cerevisiae, over a wide range of growth rate" This is a good point, as you can control the input, but the way you treat the output in your model (excluding any reaction that lead to non-carbon or complex carbon susbtrate produced as secondary metabolites or reservoir) could ocmpensate for the apparently non usable catabolic metabolic energy estimated by your model. In practice, your model is good to predict catabolic rates, but is missing important terms to account for a realistic treatment of anabolism, making conclusions on that speculative. Anyway, you must perform calculations with metabolites concentration different from standard conditions, to validate your initial model as robust, as you are assuming non-biological conditions to retrieve DeltaGs of reactions

Comments on the Quality of English Language

The English language is generally fine, but the syntax has to be improved

Author Response

(The authors gave the same response as above.)

Reviewer 4 Report

Comments and Suggestions for Authors

The text discusses the biotechnological utilization of microorganisms to produce economically valuable substances and highlights the long-standing focus on understanding microbial metabolism and its regulation. Two main approaches are outlined: one involving macrochemical equations treating microbial metabolism as a black box, and the other constructing genome-scale metabolic models by including all known reactions of an organism. Both approaches have been successful in predicting product yield.

The text acknowledges the challenge of characterizing microbial metabolism using a single equation, emphasizing the difficulty of distinguishing between anabolism and catabolism. The presented strategies aim to systematically identify separate equations for these two metabolic modes. The discussion involves the exploration of catabolic routes, their thermodynamic efficiency, and the derivation of anabolic routes to approximate biomass yield.

In conclusion, the text challenges the conventional perception of metabolism as a linear energy converter, proposing a more nuanced understanding of the relationship between catabolism and anabolism in microbial metabolism.

Author Response

(The authors gave the same response as above.)

Round 2

Reviewer 3 Report

Comments and Suggestions for Authors

Dear authors,

thanks for the excellent work of revision and for clarifying on some part of the methods.

Now it is more clear why the expected contribution of chosen concentrations would have a negligible effect on the obtained results. Moreover, the assumption on anabolic reactions is acceptable. This work represents a step forward a more comprehensive understanding of complex microbial metabolism response to external condition. Anyway, please notice that non-storage polymers produced during the formation of microbial biomass in the stationary exponential growth are more complex than what is included in the model (which is limited to 6-carbon compounds), and the inclusion of equations for more complex polymers may change a little bit the scenario in future models.

Comments on the Quality of English Language

Check any typos introduced by the extensive revision process

Author Response

Dear Reviewer,
we thank you very much for your help in improving the manuscript. We will consider non-storage polymers in future models.
We checked for typos and corrected those that we found
In the name of all authors, and best regards,
Tim Nies